# RandP: Effective and Efficient Medical Visual In-Context Learning via a Retrieve-and-Propagate Module for Prompt-Query Fusion

**Rongge Mao**[1,2]             RONGGEMAO@MAIL.USTC.EDU.CN

[1] *School of Biomedical Engineering, Division of Life Sciences and Medicine, University of Science and Technology of China (USTC), Hefei, 230026, China*

[2] *Center for Medical Imaging, Robotics, Analytic Computing & Learning (MIRACLE), Suzhou Institute for Advanced Research, USTC, Suzhou, 215123, China*

**Han Li**[*3,4]              TUM_HAN.LI@TUM.DE

[3] *Computer Aided Medical Procedures (CAMP), Technische Universitaet Muenchen (TUM).*

[4] *Munich Center for Machine Learning (MCML), Munich, Germany.*

**Chengqi Dong**[1,2]            DONGCQ@MAIL.USTC.EDU.CN

**Nassir Navab**[3,4]            NASSIR.NAVAB@TUM.DE

**S Kevin Zhou**[*1,2,5,6,7]         SKEVINZHOU@USTC.EDU.CN

[5] *Key Laboratory of Intelligent Information Processing of Chinese Academy of Sciences (CAS), Institute of Computing Technology, CAS, Beijing, 100190, China*

[6] *Jiangsu Provincial Key Laboratory of Multimodal Digital Twin Technology, Suzhou, 215123, China*

[7] *State Key Laboratory of Precision & Intelligent Chemistry, USTC, Hefei, China ORCID (S.Kevin Zhou): https://orcid.org/0000-0002-6881-4444*

**Editors:** Accepted for publication at MIDL 2026

## Abstract

Visual In-Context Learning (ICL) has emerged as a promising paradigm for constructing vision generalists by conditioning on prompt pairs. Existing visual ICL methods typically adopt a grid-like prompt-query construction combined with Masked Image Modeling (MIM) as the training strategy. However, directly applying these frameworks to medical imaging tasks often leads to suboptimal performance. Moreover, the reliance on MIM restricts the backbone to Vision Transformer (ViT) and introduces unnecessary computational overhead due to the need to reconstruct the prompt label. In this work, we revisit previous visual ICL paradigms for medical imaging and propose a training-inference aligned masking strategy to replace MIM. We further introduce a Retrieve-and-Propagate (RandP) module to enhance prompt-query fusion under this masking scheme. Experimental results show that our RandP visual ICL framework not only doubles the inference speed compared to prior visual ICL baselines but also achieves superior performance across multiple medical imaging tasks. Furthermore, unlike previous approaches constrained to vanilla ViT, our framework is compatible with U-Net-style architectures, enabling broader applicability and improved effectiveness in the medical imaging domain. Our code will be available.

**Keywords:** Medical imaging, Visual In-Context-Learning.

---

* Corresponding Authors

## 1. Introduction

Accurate medical image analysis is crucial for diagnosing various diseases (Zhou et al., 2021). With the advancement of deep learning techniques, many tasks in medical image analysis, such as classification (Yue and Li, 2024), detection (Li et al., 2022a), segmentation (Ronneberger et al., 2015), restoration (Yang et al., 2024), reconstruction (Wang et al., 2019), registration (Balakrishnan et al., 2019) and report generation (Wu et al., 2022), have made substantial progress. However, most studies concentrate on specific visual tasks, anatomical regions, and image modalities, developing specialized network architectures, training methods, and techniques tailored to these tasks. Consequently, these models often lack generalizability across different medical imaging tasks.

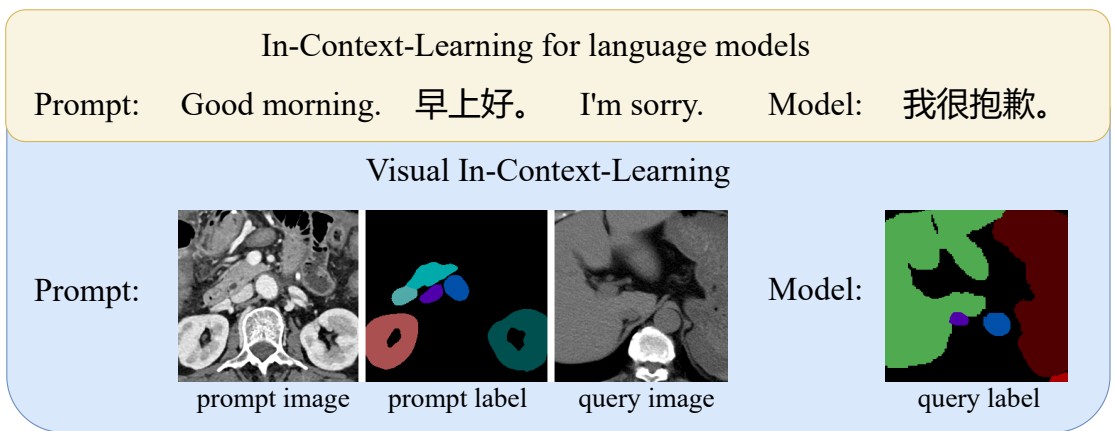

Figure 1: A concise schematic diagram of in-context learning in language models and visual in-context learning.

The impressive performance of LLMs in natural language processing (NLP) has demonstrated the potential of In-Context Learning (ICL) (Brown et al., 2020; Hao et al., 2022), a paradigm in which models perform tasks by conditioning on examples and instructions embedded in the input prompt, without any parameter updates. This framework offers several key advantages (Wei et al., 2022; Dong et al., 2024):

- **Unified Task Format:** Diverse NLP tasks can be handled under a single prompting framework, which supports new tasks through a few-shot approach and eliminates the need for task-specific fine-tuning.

- **Context-Aware Reasoning:** Task-relevant exemplars or instructions embedded in prompts guide model behavior without updating parameters.

Recent studies have demonstrated the strong effectiveness of in-context learning (ICL) for vision–language tasks, including a wide range of applications in the medical domain. In particular, studies (Zhou et al., 2024; Ferber et al., 2024) on GPT-4V (Achiam et al., 2023) show that large vision–language models can achieve competitive or even superior performance compared with expert-designed convolutional neural networks on sparse prediction

pathology tasks such as colorectal tissue typing, polyp subtyping, and lymph node metastasis detection, using as few as one to ten exemplars. Similar gains from in-context learning have also been reported for COVID-19 chest X-ray classification and tauopathy recognition tasks (Chen et al., 2023).

Consistent improvements from in-context learning have been observed across multiple medical vision–language benchmarks. Models such as Gemini Pro Vision (Team et al., 2023) and LLaVA-Med (Li et al., 2023) report gains of approximately 3–9% AUC over zero-shot baselines on radiology and pathology tasks when provided with only four to sixteen in-context examples. Moreover, medical foundation models incorporating domain-specific vocabulary and training, including PathChat (Lu et al., 2024), RadFM (Wu et al., 2023), and MUSK (Xiang et al., 2025), achieve 5-10 shots in-context learning performance that is comparable to their fully fine-tuned counterparts for medical image classification and visual question answering.

Collectively, these studies highlight the potential of in context learning as a powerful paradigm for few shot generalization in biomedical imaging, offering improved interpretability and a substantially reduced annotation burden.

However, despite these promising results, existing medical in context learning research has largely focused on sparse prediction tasks (e.g., classification, Visual Question Answering), rather than dense prediction tasks (e.g., segmentation, denoising).

Given the demonstrated effectiveness of ICL in both natural language processing (NLP) and vision-language domains for sparse prediction tasks, a natural and compelling question arises: **Can the in-context learning paradigm be effectively extended to dense prediction tasks in computer vision, particularly in the field of medical imaging?**

In the field of natural images, the answer is yes. Early visual ICL frameworks (Bar et al., 2022; Wang et al., 2023; Liu et al., 2024) for dense prediction tasks predominantly rely on Masked Image Modeling (MIM) (Xie et al., 2022; He et al., 2022) as the core training mechanism and have achieved encouraging results. Fig.1 briefly illustrates the schematic diagram of ICL in language models and visual ICL. For instance, MAE-VQGAN (Bar et al., 2022) formulates vICL as an image inpainting task by concatenating prompt-query pairs and employing a random masking strategy (He et al., 2022) to predict the discrete visual tokens (Esser et al., 2021) corresponding to the masked patches. This framework enables the model to perform various visual tasks by conditioning on different prompts. Similarly, Painter (Wang et al., 2023) adopts a simpler Masked Image Modeling approach (Xie et al., 2022), directly regressing pixel values in the image space. MVG (Ren et al., 2024) extends Painter to the medical imaging domain, adopting a hybrid training scheme that combines autoregressive training with MIM. However, as documented by (Zhou et al., 2021), medical images are characterized by reduced inter-subject variability, elevated spatial resolution, and contrast patterns that are unique to the clinical domain.

Echoing the findings of MVG (Ren et al., 2024), our experiments corroborate that directly deploying vICL pipelines designed for natural images yields sub-optimal performance on medical data, underscoring an urgent yet largely unexplored demand for domain-specific frameworks.

In this study, we identify key limitations of existing visual in-context learning models in medical imaging, such as **ineffective masking strategies, rigid backbone, and high computational overhead.** To address these challenges, we adopt a novel training-

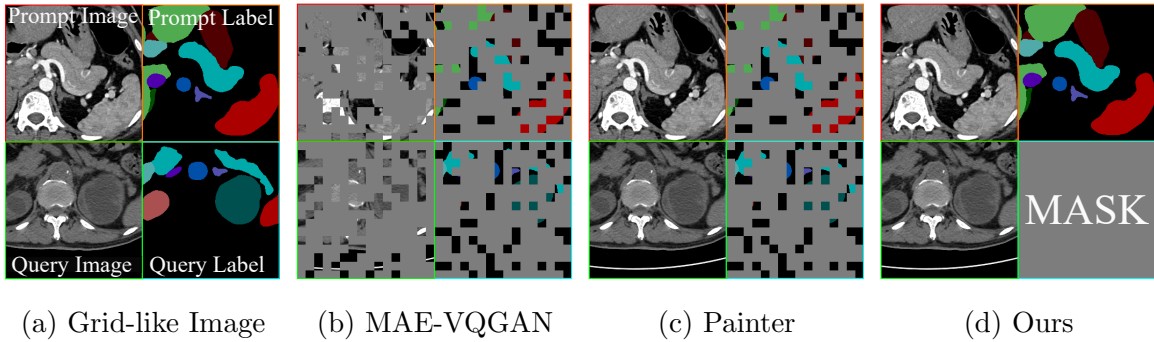

| (a) Grid-like Image | (b) MAE-VQGAN | (c) Painter | (d) Ours |

Figure 2: Illustration of different visual-ICL masking strategies during training (gray patches denote masked regions). Notably, all existing visual-ICL frameworks adopt the masking pattern shown in panel (d) at inference, whereas only our strategy preserves training–inference consistency.

inference aligned masking strategy along with a Retrieve-and-Propagate (RandP) module, both of which enhance prompt-query interaction and simultaneously reduce the number of visual tokens, which in turn improves model efficiency and performance. Unlike prior MIM-based vICL methods that are tied with ViT backbone, our framework generalizes to convolutional networks and other non-transformer designs, enabling broader applicability to dense prediction vision generalist in medical imaging. **Our main contributions are as follows**:

- We present a comprehensive analysis of the limitations of current vICL frameworks in medical imaging, regarding masking design, backbone flexibility and efficiency.

- We propose a medical vICL framework adopting a training-inference aligned masking strategy and a Retrieve-and-Propagate (RandP) module to improve performance and computational efficiency.

- Our experiments demonstrate that our framework can be effectively extended to U-Net-style (Ronneberger et al., 2015) models, which was not feasible for prior vICL frameworks.

## 2. Limitations of Previous Visual ICL Methods in Medical Imaging

Recent advances (Bar et al., 2022; Wang et al., 2023; Liu et al., 2024; Bai et al., 2024) in vICL have demonstrated strong performance across a range of vision tasks. Despite these innovations, directly applying such approaches to medical imaging remains suboptimal due to the following key limitations:

(i) **Ineffectiveness of MIM in Medical Settings.** Naive masked image modeling typically involves randomly masking a large portion (e.g., 75%) of the input image. However, in medical imaging scenarios, where images often contain large homogeneous background regions, this approach fails to preserve sufficient task-relevant information for effective prompt-query reasoning. The masking strategies shown in Fig.2.(b) and Fig.2.(c)

Table 1: Comparison of the segmentation performance, measured by the Dice Similarity Coefficient (DSC, in percentage), using different inference strategies.

|                          | MAE-VQGAN | PromptGIP | Painter |
|--------------------------|-----------|-----------|---------|
| Training-like Inference  | 83.20     | 88.64     | 86.43   |
| Modified Inference       | 1.88      | 49.09     | 23.09   |
| Standard Inference       | 44.14     | 69.92     | 78.99   |

are those adopted during the training phase of MIM-based visual ICL models.In contrast, the inference stage requires the use of a different masking strategy, as depicted in Fig.2.(d). This inconsistency results in a discrepancy between the training and inference procedures.

To further investigate this issue, we carefully design three distinct inference strategies for the trained vICL models MAE-VQGAN, Painter, and PromptGIP (Liu et al., 2024):

1. **Training-like Inference:** Matches the training phase masking strategy, providing partial visibility to the query label during inference.

2. **Modified Inference:** Replaces the partially visible query label patches with black patches before applying Training-like Inference.

3. **Standard Inference:** The typical vICL setup where the prompt image, prompt label, and query image are fully visible, and only the query label is fully masked.

Under the **Training-like Inference** setting, a small portion of the ground truth (GT) is exposed to the model. Therefore, evaluation metrics are computed only on the masked patches, without the visible GT regions to ensure a fair assessment. From Table 1, we observe that all models achieve their highest performance under the **Training-like Inference** setting. However, this setting grants the model partial access to the GT which is unrealistic in practical scenarios. When even this limited GT is removed, as in the **Modified Inference** setting, the performance drops sharply (e.g., MAE-VQGAN: $83.20\% \rightarrow 1.88\%$). These results indicate that visual ICL models trained via MIM are primarily effective at inpainting, meaning they infer missing content based on visible patches, rather than truly understanding and interpreting the query image. However, this capability is misaligned with the requirements of medical visual tasks, which demand a comprehensive understanding of the query image itself.

Moreover, comparing **Standard Inference** and **Modified Inference**, their main difference lies in whether the prompt labels are provided in full or only partially. When the prompt labels are reduced from complete to partial, model performance also declines significantly(e.g., Painter: $78.99\% \rightarrow 23.09\%$). This further underscores that prompts play a crucial role in guiding the model's processing of the query image within the vICL framework.

**(ii) Unnecessarily High Computational Overhead.** Although models like Painter and MVG attempt to mitigate this issue by adding image and label patches at shallow layers, **they still devote nearly half of their computational budget to prompt processing**. **(iii) Limited Backbone Flexibility.** MIM-based methods are tightly coupled with transformer-based architectures (Vaswani et al., 2023) and often perform poorly on convolutional networks (Tian et al., 2023), limiting their compatibility with widely used backbones in medical image analysis, such as U-Net (Ronneberger et al., 2015) and its variants, which are more suitable for image-to-image tasks in medical imaging.

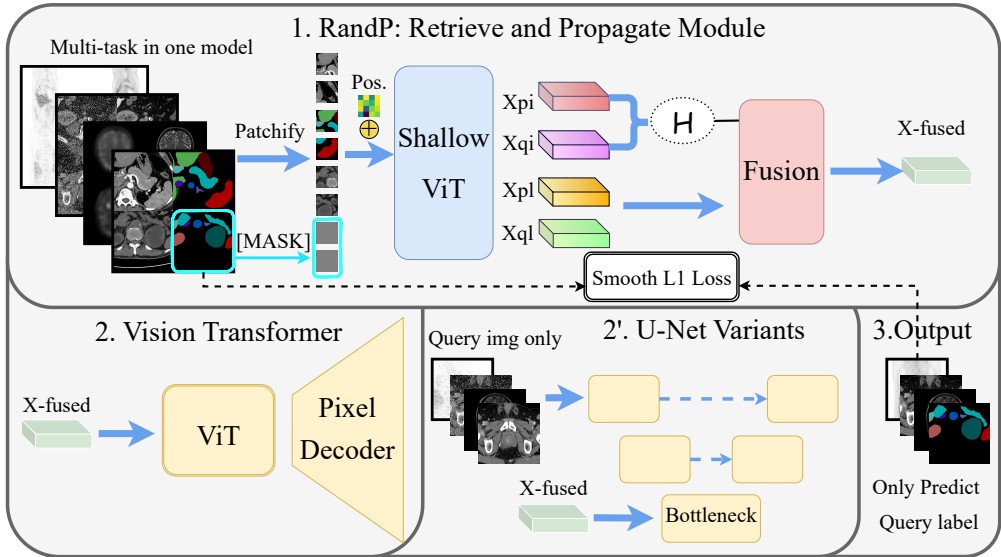

Figure 3: The grid-like image is first processed by the Retrieve-and-Propagate (RandP) Module (1) to produce *X-fused* feature, which can be fed into either a ViT backbone (2) with a pixel decoder or a U-Net-style (2') architecture to get the dense output. The query label is replaced with [MASK] tokens, which then serves as the Ground Truth to calculate the loss against the output. The italicized pi, pl, qi, and ql denote the prompt image, prompt label, query image, and query label, respectively. $H$ denotes Hungarian matching (Kuhn, 1955), and *Pos.* stands for positional encoding (Vaswani et al., 2023).

## 3. Method

**RandP Framework.** Our RandP medical vICL framework, as illustrated in the Fig.3, consists of three main components: a Retrieval-and- Propagate module for prompt-query fusion, a backbone for image feature extraction, and a pixel decoder for dense prediction.

**Unifying Input and Output Spaces.** For each *query image* in the training set, we randomly select another image with the same task and its corresponding label as the *prompt image* and *prompt label*. All four components—*prompt image*, *prompt label*, *query image*, and *query label*—are in $\mathbb{R}^{3 \times H \times W}$. These are arranged into a grid: top-left, top-right, bottom-left, and bottom-right, respectively, forming a grid-like image $X \in \mathbb{R}^{3 \times 2H \times 2W}$, as shown in Fig.2.(a). For segmentation tasks, which need to predict discrete one-hot labels, we follow (Wang et al., 2023) by assigning each semantic category a unique RGB color. During inference, the predicted label is obtained by mapping each output pixel to the nearest category via $L_2$ distance, effectively turning segmentation into an image-to-image translation task. For low-level tasks, where outputs are already continuous in RGB space, no such transformation is needed. After training, a given prompt image and label pair instructs the model to perform which task on the query image.

**Training-Inference Aligned Masking Strategy.** Unlike prior vICL approaches that predominantly rely on masked image modeling during training, we adopt a fully training-

inference aligned masking strategy, as depicted in Fig.2.(d). In our setting, the *prompt image*, *prompt label*, and *query image* are never masked, while the *query label* is always fully masked. This design offers several advantages:

1. **Consistency between training and inference:** By avoiding partial masking of the query image, the model is encouraged to truly understand the query image rather than simply reconstructing grid-like masked patterns, thus improving generalization to real ICL scenarios.

2. **Broader backbone compatibility:** This strategy allows the use of backbones beyond ViT. Prior work (Tian et al., 2023) has shown that the effectiveness of naive MIM degrades on CNNs due to the sparsity of masked inputs being diluted through stacked convolutional layers. By eliminating MIM-style masking, our framework can more effectively utilize convolutional architectures.

3. **Reduced computational cost:** Since the prompt label is never masked, there is no need to reconstruct it. Consequently, the model only needs to predict the query label, enabling us to discard prompt-related tokens after the prompt information has been integrated. This leads to significant computational savings without compromising performance.

**Retrieval-and-Propagate Module for Prompt-Query Fusion.** In previous vICL methods, prompt tokens and query tokens are concatenated into a single sequence and interact via self-attention. However, due to the quadratic complexity $\mathcal{O}(N^2)$ of self-attention with respect to sequence length, this design introduces considerable computational overhead compared to conventional image-to-image models that do not use prompts.

While the redundancy of visual tokens has been extensively validated in MLLMs (Chen et al., 2024), we argue that a similar level of redundancy may exist even when visual tokens are used purely as context. Inspired by recent efforts on visual token pruning or merging in MLLMs (Bolya et al., 2023; Zhang et al., 2025b; Wen et al., 2025), we propose to fuse prompt and query tokens at the early stage of the network to reduce computation and enhance efficiency.

We introduce a Retrieval-and-Propagate token fusion strategy, which is particularly inspired by the characteristics of medical images from different patients that tend to exhibit strong visual similarities within corresponding anatomical regions—often more pronounced than those observed in natural images. Specifically, let $\mathbf{I}_{pi}$, $\mathbf{I}_{pl}$, $\mathbf{I}_{qi}$, and $\mathbf{I}_{ql}$ denote the prompt image, prompt label, query image, and query label, respectively. These inputs are first embedded with a patch embedding layer and added with learnable positional encodings:

$$\mathbf{Z}_{pi}, \ \mathbf{Z}_{pl}, \ \mathbf{Z}_{qi} = \text{PatchEmbed}(\mathbf{I}_{pi}, \ \mathbf{I}_{pl}, \ \mathbf{I}_{qi}) + \mathbf{P} \tag{1}$$

The resulting latent representations are then fed into a shallow ViT encoder (e.g., 2 layers) to extract features:

$$\mathbf{X}_{\text{pi}}, \ \mathbf{X}_{\text{pl}}, \ \mathbf{X}_{\text{qi}}, \ \mathbf{X}_{\text{ql}} = \text{ViT}([\mathbf{Z}_{\text{pi}}, \ \mathbf{Z}_{\text{pl}}, \ \mathbf{Z}_{\text{qi}}, \ [\text{MASK}]]) \tag{2}$$

Here, $\mathbf{Z}_*$ and $\mathbf{X}_*$ represent the latent representations before and after the encoder, respectively; $\mathbf{P}$ denotes the positional encoding. [MASK] is the learnable *mask token* used to replace the masked query label.

For each patch in the query image, we compute its cosine similarity with all patches in the prompt image, effectively allowing each query token to **retrieve** similar token from the prompt. To preserve the full information of the prompt pairs, we employ Hungarian matching (Kuhn, 1955), a classical algorithm for solving optimal bipartite assignment problems, rather than greedy matching, as it enforces globally optimal one-to-one correspondences between query and prompt tokens based on the similarity matrix:

$$\mathcal{M} = \text{Hungarian}\left(-\text{Norm}(\mathbf{Z}_{\text{pi}}) \cdot \text{Norm}(\mathbf{Z}_{\text{qi}})^{\top}\right) \tag{3}$$

This design avoids information loss and ensures that each prompt token is effectively utilized in the fusion process. Although Hungarian matching has a cubic complexity with respect to the number of tokens, the token set is small in our setting, and a detailed analysis of computational complexity, runtime, and scalability is provided in the Appendix. The matched prompt and query tokens are concatenated along the feature dimension and subsequently fused via a linear layer. If a highly similar prompt token is retrieved, the model can directly reuse the associated prompt label token, effectively **propagating** it to the output.

$$\mathbf{X} = \text{Linear}\left(\text{Concat}\left[\mathcal{M}(\mathbf{X}_{\text{pi}}), \ \mathcal{M}(\mathbf{X}_{\text{pl}}), \ \mathbf{X}_{\text{qi}}, \ \mathbf{X}_{\text{ql}}\right]\right) \tag{4}$$

**Backbone and Pixel Decoder.** Similar to previous visual ICL frameworks, we use ViT as the backbone. The pixel decoder is a simple prediction head with two convolutional layers, taking the concatenated feature maps from four different layers of ViT as input (Li et al., 2022b).

**Loss Function.** The decoder outputs an image in $\mathbb{R}^{3 \times H \times W}$, and we compute the smooth L1 loss (Girshick, 2015) pixel-wise against the query label. Additionally, we use cross-entropy (CE) loss to optimize task prediction. The total loss function is as follows:

$$\mathcal{L} = \mathcal{L}_{\text{smoothL1}}(y_{\text{query\_label}}, \hat{y}_{\text{query\_label}}) + 0.1 \cdot \mathcal{L}_{\text{CE}}(y_{\text{task}}, \hat{y}_{\text{task}})$$

**Extending Visual ICL via RandP + U-Net Variants** Our masking strategy and RandP module enable the extension of visual ICL method to other non-transformer architectures, including the commonly used U-Net and its variants in medical imaging. Specifically, the activations output from the RandP Module, with a stride of 16, match the spatial dimensions of the feature maps after downsampling four times in U-Net. We adjust the channel dimensions of these activations using a $1 \times 1$ convolutional layer and add them element-wise to the feature maps in the U-Net bottleneck.

## 4. Experiment

**Dataset and Implementation.** Following (Yang et al., 2024), we select the IXI (LLC, 2024) MRI dataset for the super-resolution task, the 2016 NIH AAPM-Mayo Clinic Low-Dose CT Grand Challenge (McCollough et al., 2017) dataset for denoising task, and PET synthesis dataset provided by (Yang et al., 2024). Our segmentation dataset covers both CT and MRI modalities: PROMISE12 (Litjens et al., 2014), Prostate_MRI_Dataset (Ye et al., 2023), AMOS (Ji et al., 2022), and BTCV (Landman et al., 2015). Although we have chosen only these four tasks, our framework can be applied to any image-to-image task.

Table 2: Performance comparison across different visual ICL frameworks. Inference metrics include FLOPs, runtime (per image on a RTX 3090), and memory usage (batch size = 16).

| Models | Seg. | Denoising | | Super-Res. | | PET synthesis | | FLOPs | Time | Mem. |
|---|---|---|---|---|---|---|---|---|---|---|
| | Dice | PSNR | SSIM | PSNR | SSIM | PSNR | SSIM | (G)↓ | (ms)↓ | (GB)↓ |
| copy | 0.90 | 9.17 | 21.92 | 16.04 | 48.65 | 19.67 | 68.65 | - | - | - |
| MAE-VQGAN | 44.14 | 21.16 | 74.14 | 23.23 | 75.89 | 27.65 | 83.66 | 664 | 33.2 | 20.5 |
| PromptGIP | 69.92 | 31.1 | 90.18 | 28.36 | 88.29 | 27.77 | 86.26 | 670 | 33.1 | 18.8 |
| Painter | 78.99 | 32.83 | 91.94 | 30.11 | **91.35** | 31.16 | 89.75 | 429 | 30.6 | 15.5 |
| MVG | 82.56 | 32.97 | 92.08 | 30.11 | 91.34 | 31.09 | 89.85 | 429 | 30.5 | 15.6 |
| RandP | **84.95** | **33.01** | **92.14** | **30.14** | 91.33 | **31.46** | **90.17** | **258** | **14.4** | **10.7** |

For all models, the shallow ViT encoder in the RandP Module uses two layers. Following previous vICL frameworks (Wang et al., 2023; Ren et al., 2024; Liu et al., 2024), we adopt the identical experimental protocol, wherein the prompt–query fusion module, the backbone network, and the pixel decoder are all trained in an end-to-end manner. We train for 100 epochs with a maximum learning rate of $1e-3$, a batch size of 64, and a Grid-like Image resolution of $512 \times 512$. In this paper, the reported Dice (Dice, 1945) and SSIM (Wang et al., 2004) values are presented in percentage form.

**Comparison of Different Visual ICL Frameworks.** We re-implement multiple prior vICL methods on our medical datasets, including MAE-VQGAN, PromptGIP, Painter, and MVG, to construct strong baselines for comparison. Additionally, we include AMIR (Yang et al., 2024), a router-based multi-task model in medical imaging. We also introduce a **copy** baseline, which simply replicates the prompt label as the final output. For MAE-VQGAN, a pretrained VQGAN (Esser et al., 2021) is required to serve as the tokenizer. We initialized the VQGAN with weights pretrained on ImageNet (Deng et al., 2009) and further fine-

Table 3: Performance comparison under the Single-Task Separate Training Setting.

| Models | Seg. | Denoising | | Super-Resolution | | PET synthesis | |
|---|---|---|---|---|---|---|---|
| | Dice | PSNR | SSIM | PSNR | SSIM | PSNR | SSIM |
| AMIR (Yang et al., 2024) | 84.45 | 33.71 | 92.47 | 30.52 | 92.11 | **31.56** | **90.48** |
| MVG (Ren et al., 2024) | 77.91 | 32.54 | 91.65 | 29.94 | 91.08 | 31.19 | 89.96 |
| RandP | 79.32 | 32.80 | 91.88 | 29.93 | 91.17 | 31.18 | 89.97 |
| RandP-UX-NET | 83.27 | 32.77 | 89.82 | 29.44 | 91.02 | 30.40 | 88.72 |
| RandP-SwinUNETR | 83.63 | 32.44 | 89.33 | 29.16 | 90.25 | 30.67 | 89.13 |
| RandP-Restormer | **84.89** | **33.73** | **92.58** | **30.54** | **92.27** | 31.21 | 90.14 |

Table 4: Performance comparison under the Multi-Task Joint Training Setting. Values in parentheses indicate the performance gain compared to single-task training. UX., Sw., and Res. refer to UX-NET, SwinUNETR, and Restormer, respectively.

| Models | Segment. | Denoising | | Super-Resolution | | PET synthesis | |
|---|---|---|---|---|---|---|---|
| | Dice | PSNR | SSIM | PSNR | SSIM | PSNR | SSIM |
| AMIR | 83.5(-0.9) | 33.9(+0.2) | 92.8(+0.3) | 30.8(+0.3) | 92.3(+0.2) | 31.6(+0.1) | 90.6(+0.1) |
| RandP | 85.0(+5.6) | 33.0(+0.2) | 92.1(+0.2) | 30.1(+0.2) | 91.3(+0.2) | 31.5(+0.3) | 90.2(+0.2) |
| RandP-UX. | 84.1(+0.8) | 33.8(+1.0) | 92.7(+2.9) | 30.5(+1.0) | 92.0(+1.0) | 31.6(+1.2) | 90.4(+1.6) |
| RandP-Sw. | 84.2(+0.6) | 33.7(+1.3) | 92.6(+3.3) | 30.3(+1.1) | 91.6(+1.4) | 31.6(+0.9) | 90.3(+1.1) |
| RandP-Res. | 85.7(+0.8) | 34.0(+0.3) | 92.9(+0.3) | 30.8(+0.3) | 92.5(+0.2) | 32.1(+0.9) | 91.0(+0.9) |

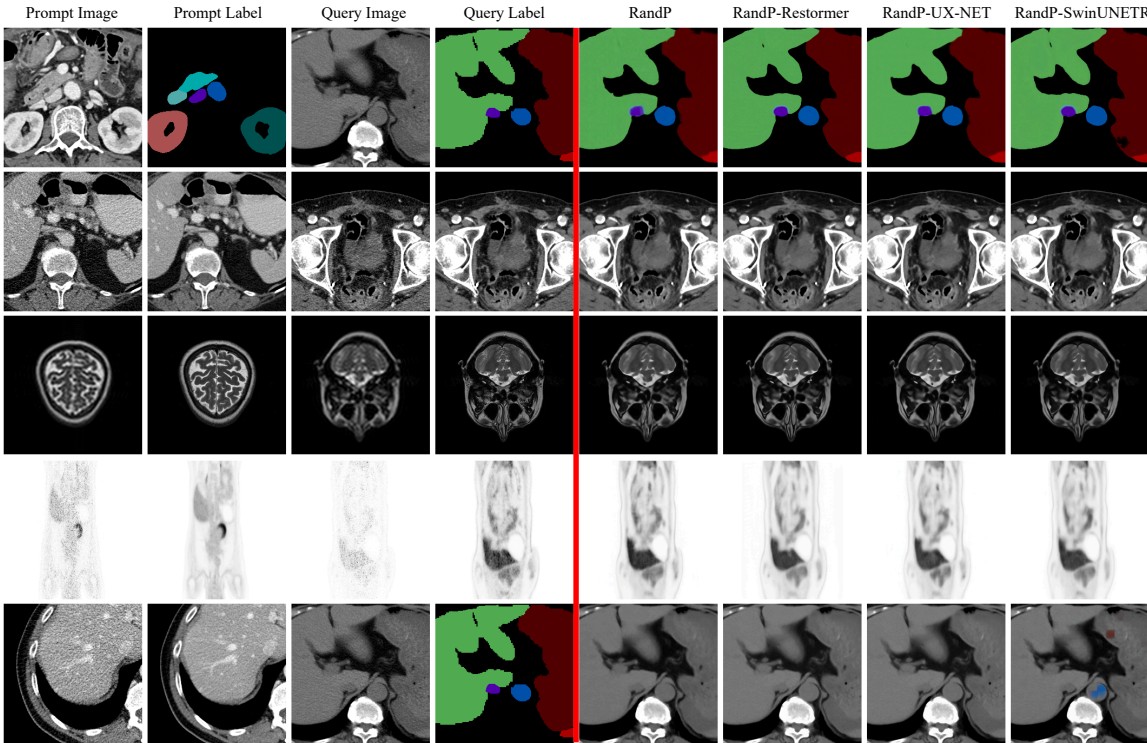

Figure 4: Qualitative evaluation of our RandP models.

tuned it on our datasets. In contrast, all other models perform regression directly in the pixel space. All frameworks adopt ViT as the backbone. As shown in Table 2, our proposed **RandP** framework consistently outperforms previous vICL frameworks across various tasks, while achieving **inference speed 2× faster** than prior visual ICL frameworks and **lower memory consumption**.

**Extending RandP to U-Net Variants.** We extend the RandP framework to several widely-used U-Net variants in medical imaging. Specifically, we adopt pure advanced convolutional UX-Net (Lee et al., 2023), and SwinUNETR (Hatamizadeh et al., 2022), a window attention (Liu et al., 2021) based model. Both of them originally designed for 3D tasks, we adapt them to 2D settings. Additionally, we incorporate Restormer (Zamir et al., 2022), a modified-transformer-based model commonly used for low-level vision tasks. These combinations result in three RandP-based medical vICL models: RandP-UX-Net, RandP-SwinUNETR, and RandP-Restormer. We first train these models independently for each task. As shown in Table 3, in the single-task separate training setting, models with the same backbone generally show similar performance—for example, RandP and MVG (with ViT), and RandP-Restormer and AMIR (with Restormer) perform comparably. However, ViT-based models still lag behind task-specific U-Net-style architectures, particularly for the segmentation task.

We further perform joint multi-task training using the RandP framework across different backbones. As shown in Table 4, compared to single-task training, all backbones consistently benefit from joint optimization, demonstrating RandP's capability to mitigate

inter-task conflicts during learning. However, the router-based AMIR framework shows a performance drop on the segmentation task under multi-task training compared to its single-task counterpart.

Both RandP-Restormer and AMIR use Restormer as the backbone. While AMIR introduces a complex task routing mechanism and incurs additional computational cost to reduce task interference, RandP-Restormer achieves clear and consistent improvements across all tasks, highlighting the effectiveness and efficiency of the RandP framework.

Fig. 4 shows the qualitative evaluation of our RandP models, the last row of it indicates that when we provide a denoising prompt for a segmentation query image, the model executes the task as instructed by the prompt, rather than simply memorizing the dataset.

Table 5: Performance difference between random prompts, learned prompts and selected prompt

|  | Segmentation | Denoising | | Super-Resolution | | PET synthesis | |
|---|---|---|---|---|---|---|---|
|  | Dice | PSNR | SSIM | PSNR | SSIM | PSNR | SSIM |
| **Random Prompt** | 84.95 | 33.01 | 92.14 | 30.14 | 91.33 | 31.46 | 90.17 |
| **Learned Prompt** | 85.11 | 33.04 | 92.16 | 30.12 | 91.32 | 31.39 | 90.16 |
| **Selected Prompt** | **85.56** | **33.29** | **92.50** | **30.21** | **91.49** | **32.31** | **91.32** |

**Random Prompt vs. Learned Prompt vs. Selected Prompt** In addition, we froze the parameters of the trained visual ICL model and treated the prompt image and prompt label as learnable embeddings. Each task was associated with a distinct set of prompt embeddings, enabling task-specific adaptation without modifying the backbone. These learnable embeddings are optimized using a learning rate of 1e-4 for 10 epochs without any warm-up schedule. During inference, we used the corresponding learned prompt embeddings for each task. As shown in Table 5, due to the marginal performance gap between learned and random prompts, the use of learned prompts can be seen to enhance the stability of medical visual ICL models. Upon visualizing the learned prompts, we find that they do not resemble semantically meaningful images; rather, they appear as structured noise patterns with no obvious visual interpretation. Building on the prompt-selection protocol proposed by  (Ferber et al., 2024; Zhang et al., 2023), we retained the most relevant prompt via an external visual encoder;; the resulting "Selected Prompt" row in Table 8 demonstrates a clear performance gain. Sophisticated prompt-selection strategies (Sun et al., 2025; Suo et al., 2024) have recently been introduced in natural-image vICL, and we consider their adaptation to medical imaging a promising direction for future research.

**Ablation Study 1: Query-Prompt Interaction strategy.** Painter and MVG adopt apatch merging mechanism to fuse images and labels. However, as shown in Table 6, under the training-inference aligned masking strategy and the setting where the prompt label is not reconstructed, the performance of patch merging (i.e., spatially aligned visual tokens from the prompt image, prompt label, query image, and query label are summed at the merge layer) is significantly inferior to that of our proposed RandP Module. Another important aspect concerns the matching strategy between prompt tokens and query tokens during the fusion process. We experimented with both greedy matching and Hungarian matching. The primary difference lies in whether a single prompt token is allowed to match multiple query tokens. Experimental results show that Hungarian matching significantly

Table 6: Ablation Study Results. Dice and SSIM are reported in percentage.

| | Seg. | Denoising | | Super-Resolution | | PET synthesis | |
|---|---|---|---|---|---|---|---|
| **Models** | Dice | PSNR | SSIM | PSNR | SSIM | PSNR | SSIM |
| **prompt-query token fusion strategies in ViT backbone** | | | | | | | |
| Patch Merge | 75.91 | 32.94 | 92.09 | 29.99 | 90.95 | 31.25 | 90.09 |
| Greedy Matching | 78.83 | 32.99 | 92.09 | 29.38 | 90.11 | 31.03 | 89.53 |
| Hungarian Matching | **84.95** | **33.01** | **92.14** | **30.14** | **91.33** | **31.46** | **90.17** |
| **Different Fusion Strategies when extending RandP to Restormer backbone** | | | | | | | |
| First-Stage | 84.71 | 33.97 | 92.89 | 30.74 | 92.31 | 31.44 | 90.38 |
| Multi-Stage | 83.62 | 33.97 | 92.78 | **30.86** | 92.47 | 31.13 | 90.12 |
| Bottleneck | **85.70** | **33.99** | **92.92** | 30.83 | **92.50** | **32.08** | **91.04** |

outperforms greedy matching. We hypothesize that many-to-one matching leads to the neglect of numerous prompt tokens, thereby causing substantial prompt information loss.

**Ablation Study 2: Fusion Strategy for Extending RandP to U-Net Variants.** The mixed features from the RandP module are $16\times$ downsampled relative to the input image. We explore three fusion strategies to integrate them into U-Net-style architectures:

- **First-stage fusion:** Upsample the mixed features via pixel shuffle and concatenate with the input image at the first encoder stage.

- **Multi-stage fusion:** Upsample the mixed features to multiple scales and add them to encoder features at corresponding stages.

- **Bottleneck fusion:** Inject the mixed features directly into the bottleneck layer of the encoder.

Experimental results show that bottleneck fusion achieves the best performance. We hypothesize that this is because early stages of U-Net primarily focus on low-level features such as textures and edges (Horwath et al., 2020), where introducing prompt-related information provides limited benefit. In contrast, the role of prompts is more aligned with guiding high-level semantic understanding, for example by indicating the task being performed (Hojel et al., 2024) and providing task-specific insights (Han et al., 2025). Injecting prompt-related information at fine spatial resolutions may therefore be suboptimal, as the prompt inevitably differs from the query image in low-level details. Since the model output is solely optimized for understanding the query image, introducing fine-grained prompt features at early stages may even be detrimental to query interpretation.

## 5. Conclusion

In this paper, we propose a medical vICL framework called RandP, which enables the execution of multiple different medical imaging tasks via visual prompt pairs. Our experiments demonstrate that RandP has superior performance while maintaining low computational cost. Furthermore, RandP can be extended to other architectures beyond ViT.

## Acknowledgments

This work was supported by Natural Science Foundation of China under Grant 62271465, Suzhou Basic Research Program under Grant SYG202338, and Open Fund Project of Guangdong Academy of Medical Sciences, China (No. YKY-KF202206).

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

## Appendix A. Hungarian Matching for Prompt–Query Fusion

### A.1. Problem Formulation

Hungarian matching (Kuhn, 1955) is a classical algorithm for solving the minimum-cost bipartite assignment problem. Given a cost matrix $\mathbf{C} \in \mathbb{R}^{N \times N}$, it computes a globally optimal one-to-one assignment between two sets by minimizing the total matching cost. In our framework, Hungarian matching is applied to align a small set of prompt tokens with query tokens, enforcing explicit and globally consistent correspondences.

### A.2. Computational Complexity and Runtime Analysis

Due to the $\mathcal{O}(N^3)$ complexity of Hungarian matching, it is generally unsuitable for large-scale dense token matching. However, in our design, the algorithm is intentionally restricted to a small set of prompt–query tokens rather than dense patch tokens, ensuring that the computational overhead remains manageable in practice.

We measured the execution times for the Hungarian matching step using both the Scipy implementation and the GPU-accelerated Hungarian matching algorithm from the HA4DeTR (Feng Lin, 2025) for varying token sequence lengths. Table 7 reports the per-matrix runtime measured on an NVIDIA RTX 3090 GPU.

When $N \leq 256$, which is the setting used throughout all experiments in this work, the GPU overhead of Hungarian matching is below 0.2 ms and remains negligible compared to the forward pass of a ViT backbone (approximately 13.8 ms). As expected, the runtime increases cubically with respect to $N$, and Hungarian matching becomes the computational bottleneck for larger token sets (e.g., $N \geq 1024$).

### A.3. Scalability Considerations

The above results indicate that Hungarian matching does not scale favorably to large numbers of tokens. This limitation is intrinsic to the algorithm's cubic complexity and motivates our design choice to restrict its usage to small, semantically meaningful prompt–query sets.

For scenarios involving higher resolution, several extensions are possible:

- **Windowed matching**, where Hungarian matching is applied only within local spatial windows to exploit locality and reduce computational cost.

- **Approximate assignment algorithms**, such as the Sinkhorn algorithm (Sinkhorn and Knopp, 1967), as an alternative to Hungarian matching.

Table 7: Runtime of Hungarian matching for different token sizes.

| Number of tokens | GPU (ms) | SciPy (ms) |
|---|---|---|
| 256 | 0.125 | 0.57 |
| 512 | 0.427 | 3.05 |
| 1024 | 1.607 | 12.93 |
| 2048 | 8.082 | 54.38 |

Our goal in this work is not to scale Hungarian matching to large token sets, but to study the effectiveness of globally optimal one-to-one alignment for structured prompt–query fusion. Within this scope, Hungarian matching provides a principled and reliable mechanism for enforcing explicit correspondence, while its computational cost remains well controlled. We leave the exploration of these extensions to future work.

## Appendix B. Prompt Analysis

Numerous studies (Bar et al., 2022; Sun et al., 2025; Zhang et al., 2023) have highlighted the sensitivity of vICL models to the choice of prompts. Some works have explored heuristic strategies (Sun et al., 2025; Zhang et al., 2023; Suo et al., 2024) for selecting optimal visual prompts, while others have investigated learning-based approaches (Jia et al., 2022; Zhang et al., 2024, 2025a) to prompt construction. Given the high demand for controllability in medical image analysis, vICL models are designed to take both the query image and a prompt pair—consisting of an image and its corresponding label—as input. Therefore, we conduct a comprehensive analysis of the prompt component in trained visual ICL models.

Table 8: Average standard deviation of the performance of trained medical vICL models across 20 different prompts.

|  | Segmentation | Denoising | Super-Resolution | PET synthesis |
|---|---|---|---|---|
|  | Dice-std | | PSNR-std | |
| **MVG** | **0.00455** | 0.00880 | **0.0057** | 0.0353 |
| **RandP** | 0.00704 | 0.00695 | 0.0069 | 0.0416 |
| **RandP-UX-NET** | 0.00855 | 0.00526 | 0.0195 | 0.0327 |
| **RandP-SwinUNETR** | 0.00968 | 0.00815 | 0.0272 | **0.0284** |
| **RandP-Restormer** | 0.00722 | **0.00475** | 0.0239 | 0.0336 |

**Standard Deviations Across Different Prompt Pairs.** First, for each query image in the test set, 20 prompt pairs from the same task were randomly selected for inference. We then calculated the standard deviation of performance metrics across these inferences. Finally, we averaged these standard deviations over the entire test set. The results are summarized in Table 8.

When using ViT as the backbone, our RandP model exhibits higher standard deviations in segmentation, super-resolution, and PET synthesis compared to MVG. We attribute this to our more aggressive merging strategy between prompt tokens and query tokens. When extending the RandP framework to other U-Net variants, the standard deviation tends to increase further across most tasks relative to the ViT-based version.

However, a lower standard deviation is not always preferable. We argue that a moderately low standard deviation is desirable—it ensures model stability across prompts while allowing prompts to exert meaningful influence on the model's interpretation of the query. The results suggest that our trained medical visual ICL models under the RandP framework maintain appropriately low standard deviations, ensuring stable performance under different prompts. Meanwhile, the variability in prompt effectiveness implies that some prompts are better than others. Identifying optimal prompts for medical visual ICL models will be an important direction for our future work.

## Appendix C. Pseudocode of RandP

---

**Algorithm 1** Retrieval and Propagate for Prompt-Query Interaction

---

**Input:** Prompt image $\mathbf{I}_{pi}$, Prompt label $\mathbf{I}_{pl}$, Query image $\mathbf{I}_{qi}$

**Output:** Prompt-Query Fusion Feature $\mathbf{X}$

**Step 1: Patch Embedding**

$\quad \mathbf{Z}_{pi}, \mathbf{Z}_{pl}, \mathbf{Z}_{qi} \leftarrow \text{PatchEmbed}(\mathbf{I}_{pi}, \mathbf{I}_{pl}, \mathbf{I}_{qi}) + \mathbf{P}$

**Step 2: Shallow ViT for Feature Extraction**

$\quad \mathbf{X}_{pi}, \mathbf{X}_{pl}, \mathbf{X}_{qi}, \mathbf{X}_{ql} \leftarrow \text{ViT}([\mathbf{Z}_{pi}, \mathbf{Z}_{pl}, \mathbf{Z}_{qi}, [\text{MASK}]])$

**Step 3: Cosine Similarity Matching**

$\quad \mathbf{S} \leftarrow \text{Norm}(\mathbf{X}_{pi}) \cdot \text{Norm}(\mathbf{X}_{qi})^{\top}$

$\quad \mathcal{M} \leftarrow \text{Hungarian}(-\mathbf{S})$

**Step 4: Reorder Prompt Tokens**

$\quad \mathcal{M}(\mathbf{X}_{pi}) \leftarrow \text{Reorder}(\mathbf{X}_{pi}, \mathcal{M})$

$\quad \mathcal{M}(\mathbf{X}_{pl}) \leftarrow \text{Reorder}(\mathbf{X}_{pl}, \mathcal{M})$

**Step 5: Prompt-Query Token Fusion**

$\quad \mathbf{X} \leftarrow \text{Linear}(\text{Concat}[\mathcal{M}(\mathbf{X}_{pi}), \mathcal{M}(\mathbf{X}_{pl}), \mathbf{X}_{qi}, \mathbf{X}_{ql}])$

**Return X**

---

