# OpenReview forum: "RandP: Effective and Efficient Medical Visual In-Context Learning via a Retrieve-and-Propagate Module for Prompt-Query Fusion"
_MIDL.io/2026/Conference — MIDL 2026 Poster_

### Official Review · Reviewer_tMMC · 2025-12-22

**Confidence:** 4
**Preliminary Rating:** 4
**Final Rating:** 5

**Summary:**

The paper introduces RandP, a framework for medical Visual In-Context Learning (VICL) designed for dense prediction tasks. It replaces traditional Masked Image Modeling (MIM) with a training-inference aligned masking strategy and a Retrieve-and-Propagate (RandP) module to improve efficiency and performance. Unlike previous methods, RandP is compatible with both Vision Transformers and U-Net-style architectures.

**Strengths:**

1. Achieves inference speeds 2x faster than prior baselines with significantly lower memory consumption.
2. Successfully extends VICL to non-transformer backbones like UX-Net, SwinUNETR, and Restormer.
3. Eliminates the training-test discrepancy found in MIM-based methods by using a consistent masking scheme.

**Weaknesses:**

1. Evaluation is restricted to four tasks; more complex medical imaging tasks like registration are mentioned but not tested.
2. By injecting fused features exclusively at the U-Net bottleneck, the model might bypass the high-resolution spatial details typically preserved by skip connections. This could limit the precision of fine-grained boundaries in complex dense prediction tasks like micro-segmentation.

**Detailed Comments:**

1. While the model achieves a 2x speedup , the paper should explicitly quantify how much of this is due to the shallow 2-layer ViT encoder versus the decision to discard prompt tokens after the fusion stage.
2. Although Hungarian matching significantly improves performance over greedy matching, it typically has higher algorithmic complexity. A brief discussion on how this affects training time compared to standard self-attention would be beneficial.
3. The authors found that bottleneck fusion is superior to multi-stage fusion. Clarification is needed on whether the high-resolution skip connections in U-Net variants (like UX-Net) still function normally or if the injected RandP features at the bottleneck dominate the decoding process.

**Justification Of Final Rating:**

Thank you for providing such a comprehensive and detailed response to my previous inquiries. Your explanation has mostly addressed the majority of my concerns. I appreciate the additional context you provided regarding the experimental results.

**Justification Of The Preliminary Rating:**

The framework has clear motivation to resolve the "inpainting shortcut" flaw in prior models. The paper is highly polished, featuring intuitive diagrams like Figure 1, which clarifies masking differences, and Figure 2, which maps the fusion process for both ViT and U-Net backbones. Empirically, the model achieves a 2x inference speedup and superior performance across all medical tasks as evidenced in Table 2.

**Questions To Address In The Rebuttal:**

See weakness.

---

> ### Author Response · Authors · 2026-01-23
>
> We appreciate the review for pointing out some issues that require in-depth discussion and providing valuable opinions.
>
> Comment1: More complex medical imaging tasks like registration are mentioned but not tested.
>
> Response1: We have fine-tuned our pre-trained RandP model for 20 epochs on the Neurite-OASIS[1] dataset registration task to preliminarily validate the effectiveness of our method for this task. The results are shown in the following table.
>
> | Dataset       | Model | Dice   |
> |---------------|-------|--------|
> | Neurite-OASIS | RandP | 62.51% |
>
> Registering such complex tasks may require additional design to better transform them into an img-to-img mode and thus integrate them into the vICL framework. We will conduct more detailed research in our future work.
>
> Comment2: "While the model achieves a 2x speedup, the paper should explicitly quantify how much of this is due to the shallow 2-layer ViT encoder versus the decision to discard prompt tokens after the fusion stage."
>
> Response2:  The following table lists the number of tokens for different models at various stages, providing a clearer depiction of why RandP is more efficient:
>
> | Seq_length | Painter/MVG | RandP |
> |------------|-------------|-------|
> | Layer0 -1  | 1024        | 1024  |
> | Layer2-23  | 512         | 256   |
>
> Comment3: "Although Hungarian matching significantly improves performance over greedy matching, it typically has higher algorithmic complexity. A brief discussion on how this affects training time compared to standard self-attention would be beneficial." We appreciate you pointing out this key issue."
>
> Response3: We have added an introduction to Hungarian matching, along with a discussion and analysis of its computational complexity, specific running times, and scalability, in the appendix. We list the execution times for the Hungarian matching step using both the Sicpy implementation and the GPU-accelerated Hungarian matching algorithm from the HA4DeTR[2] for varying token sequence lengths in the table below.
>
> | Seq_length | GPU (ms) | SciPy (ms) | ViT (ms) |
> |------------|----------|------------|----------|
> | 256        | 0.125    | 0.57       | 13.8     |
> | 512        | 0.427    | 3.05       | —        |
> | 1024       | 1.607    | 12.93      | —        |
> | 2048       | 8.082    | 54.38      | —        |
>
> Comment4: "The authors found that bottleneck fusion is superior to multi-stage fusion. Clarification is needed on whether the high-resolution skip connections in U-Net variants (like UX-Net) still function normally or if the injected RandP features at the bottleneck dominate the decoding process." And "By injecting fused features exclusively at the U-Net bottleneck, the model might bypass the high-resolution spatial details typically preserved by skip connections. This could limit the precision of fine-grained boundaries in complex dense prediction tasks like micro-segmentation."
>
> Response4: It should be clarified that only the query image undergoes downsampling via the U-Net encoder and fine-grained skip connections for detail preservation. Prompt information is injected at the bottleneck, a level more focused on semantics. Given the inevitable differences in details between the prompt image and the query image, we believe it is inappropriate to inject prompt information at high-resolution features. This is also supported by the better performance of bottleneck fusion compared to multi-stage fusion and first-stage fusion in our ablation experiments. Meanwhile, for the query image, fine-grained boundaries in complex dense prediction tasks are still preserved through skip connections. In the revised version, we have also supplemented a discussion regarding why bottleneck fusion achieves the best performance.
>
> [1]Marcus, Daniel S., et al. "Open Access Series of Imaging Studies (OASIS): cross-sectional MRI data in young, middle aged, nondemented, and demented older adults." Journal of cognitive neuroscience 19.9 (2007): 1498-1507.
>
> [2]Feng Lin, Xiaotian Yu, and Rong Xiao. GPU-Accelerated Batched Hungarian Algorithm for DETR. GitHub repository, 2025. https://github.com/linfeng93/HA4DETR

---

### Official Review · Reviewer_NsvT · 2026-01-01

**Confidence:** 5
**Preliminary Rating:** 3
**Final Rating:** 5

**Summary:**

The paper proposes an interesting solution to simultaneously address computational efficiency, architectural flexibility as well as modeling efficacy for imaging-to-imaging tasks in medical scenarios through visual in-context learning. The proposed Retrieve-and-Propagate (RandP) module effectively summarizes the prompt and query images along with prompt labels into a latent representation, from which new images, i.e., query labels, can be generated or predicted.

**Strengths:**

The proposed method’s intuitivity, effectiveness and gains on time and space requirements are impressive. Its usability with both ViTs and CNNs, especially well-established U-Net variants, makes it a potential plug-and-play solution for many image analysis tools out there. Experimental results also support the claims made in the text.

**Weaknesses:**

Despite the strengths above, the manuscript would substantially benefit from refinements and restructuring, especially in earlier sections. Section 1 and 2 should provide a clear introduction and sufficient background to the reader, especially considering the wide range of audience in the medical imaging community. However, it seems like the reader was assumed to know many things beforehand. I see that the authors as experts are in command of technical details but a little extra effort needs to be made to communicate things better to the reader. I will list my recommendations below in the detailed comments section.

**Detailed Comments:**

1. Introduction:
In-context learning has attracted a lot of attention recently; however, it would be nice to have a brief description of it along with that of prompt, query and their interactions. Between the 1st and 2nd paragraphs would be a nice place to do this. Once these are given, they can be used to describe grid-like images, which are essential to RandP. Such background information will also be useful for understanding references to related work in later paragraphs. On top of that, please, use the full form of acronyms before switching to shorthand notation. ICL appears before In-Context Learning (ICL) in text.

Is ICL possible only with GPT-4V? No. There are other ICL examples in medicine based on Gemini, etc.. It would be good to diversify the examples of ICL.

If the pioneering studies rely on masked image modeling (MIM), explicitly say it first, use the acronym MIM, and then provide examples of related work. While this is partially done, a better structure is needed.

“However, the distinct properties of medical images limit the effectiveness of directly adopting natural image vICL methods, revealing a critical yet underexplored need for domain-specific frameworks.” – Is this a fact or speculation? Do you have any reference or evidence to support this sentence?

The contribution list does not need the titles in italic. The same information is already given in the following sentences anyway.

2. Limitations of Previous vICL Methods in Medical Imaging

“Furthermore, as illustrated in Fig.1.(a) and Fig.1.(c), the masking strategies shown are those adopted during the training phase of MIM-based visual ICL models.” – I read this sentence multiple times to make sense of it. It is a bit broken, I guess. Maybe, rewrite it as “The masking strategies shown in Fig.1.(a) and Fig.1.(c) are those adopted during the training phase of MIM-based visual ICL models.”

“In contrast, the inference stage requires the use of a different masking strategy, as depicted in Fig.1.(d).” – If Fig.1.d is about inference, why does the caption of Fig.1 say it is about training? Please, adjust the figure caption and text accordingly.

“This inconsistency results in a discrepancy between the training and inference procedures, which introduces a gap between training and inference.” – This sentence is also repeating the same thing twice. Yes, there is a discrepancy between training and inference and the paragraph could be refined into a shorter and more effective one.

Standard inference, training-like inference and modified inference:
If modified inference is the same as training-like inference but with query label patches replaced with black patches, does it in fact mean “no visible information”? Or if it is as described later (Standard inference vs Modified Inference), please move that description closer to definition. The reader should not search for parts across the manuscript. Also, be consistent with the ordering of items (1,2,3) and their discussion/explanation.

“ii) Unnecessarily High Computational Overhead. Although models like Painter and MVG attempt to mitigate this issue by adding image and label patches at shallow layers, they still devote nearly half of their computational budget to prompt processing. This allocation limits the model’s capacity to focus on the query task itself, reducing overall efficiency.” – Who established this? Is there a reference to use here?

3. Method

Retrieval-and-Propagate Module for Prompt-Query Fusion:
“We introduce a Retrieval-and-Propagate token fusion strategy, which is particularly inspired by the characteristics of medical images from different patients “that” tend to exhibit strong visual similarities within corresponding anatomical regions” – I believe we need “that” or something to bridge two ends there.

Retrieve and Propagating: do we really need capital letters for these? A bold face should be enough to highlight these keywords used for the proposed method. Same goes for “Copy” baseline.

What is the Hungarian matching? It needs a little explanation (basic definition and implications). Also, what is the complexity of the Hungarian matching algorithm used here? How does it compare to attention? Is it worth it to abolish one or the other?

4. Experiments

By the definition of ICL, we should not change the backbone model’s weights too much. It will be pretty much untouched while we learn from the given context set forth through examples and prompts. So, I assume only the shallow ViT and linear parts of the RandP module were trained for 100 epochs and the rest (large ViT, pixel decoder, etc..) was frozen. If this is the case, it should be made more explicit. Otherwise, would we still call it ICL? If we finetune the whole stack, it would be the old school supervised learning, no?

Table 5 shows First-stage, Multi-stage and Bottleneck; however, in the text, they are given as First, Bottleneck and Multi. Please, be consistent.

**Justification Of Final Rating:**

The authors have diligently addressed all of my comments. The manuscript is quite informative and inspiring now.
I am happy to change my preliminary rating from "borderline" to "strong accept" in the light of substantial improvements and potential of RandP for many extensions and downstream applications. With that said, an extended version of the work could be considered for MELBA Special Issue.

I would love to see this work at MIDL 2026.

**Justification Of The Preliminary Rating:**

Despite the potential of RandP, the manuscript seems to have been rushed to beat the deadline without sufficient rounds of internal reviews. Senior authors should have provided more oversight to avoid those issues in writing and presentation. Such supervisory tasks should not be delegated to reviewers. If they are addressed properly during rebuttal, I will be happy to adjust my rating accordingly.

**Questions To Address In The Rebuttal:**

In addition to the detailed comments above, please, see the followings:

3. Method

“For each query image in the training set, we randomly select another image with the same task and its corresponding label as the prompt image and prompt label.”

Random selection of images is an easy but usually a good starting point. As per Appendix, learned prompts were also tested but did not lead to significant improvements. While learned prompts might require further investigation and hyperparameter tuning, could you try a simple kNN-based selection? In the works of Ferber et al (2024) in pathology as well as others in ophthalmology, kNN-based selection worked quite well. It would be nice to see its impact here.

4. Experiments

“Experiments show that bottleneck fusion achieves the best performance.” – Yes, it is what we see from the table. Could you elaborate on why this is the case? Maybe, provide a little discussion or speculation to somehow make the reader contemplate or gain insights.

---

> ### Author Response · Authors · 2026-01-23
> **Introduction**
>
> We sincerely appreciate the reviewer's invaluable and insightful comments.
>
> Comment1: "Introduction: In-context learning has attracted a lot of attention recently; however, it would be nice to have a brief description of it along with that of prompt, query and their interactions. Between the 1st and 2nd paragraphs would be a nice place to do this. Once these are given, they can be used to describe grid-like images, which are essential to RandP. Such background information will also be useful for understanding references to related work in later paragraphs. On top of that, please, use the full form of acronyms before switching to shorthand notation. ICL appears before In-Context Learning (ICL) in text."
>
> Response1: In the revised introduction, we have added an extra introduction to ICL in the Introduction section, including its advantages. We have also incorporated Figure 1 to illustrate text-based ICL and visual ICL, facilitating the introduction of the roles of prompt image, prompt label, query image, and query label within the vICL framework. Additionally, we have systematically corrected acronym usage throughout the manuscript
>
> Comment2: "Is ICL possible only with GPT-4V? No. There are other ICL examples in medicine based on Gemini, etc.. It would be good to diversify the examples of ICL."
>
> Response2: We have included more and broader examples of effective medical vICL in the Introduction section.
>
> Comment3:  "If the pioneering studies rely on masked image modeling (MIM), explicitly say it first, use the acronym MIM, and then provide examples of related work. While this is partially done, a better structure is needed."
>
> Response3: In the revised version, we have ensured that the acronym MIM (Masked Image Modeling) is explicitly introduced upon its first mention. We have also restructured the narrative to first clarify the reliance of previous vICL frameworks on MIM, followed by a systematic presentation of relevant studies as examples. This adjustment enhances clarity and logical flow while adhering to academic writing conventions.
>
> Comment4:  "However, the distinct properties of medical images limit the effectiveness of directly adopting natural image vICL methods, revealing a critical yet underexplored need for domain-specific frameworks.” – Is this a fact or speculation? Do you have any reference or evidence to support this sentence?"
>
> Response4: We have listed specific differences between medical images and natural images from the literature and added citations to  underscore the necessity of designing domain-specific frameworks for medical vICL.
>
> Comment5: "The contribution list does not need the titles in italic. The same information is already given in the following sentences anyway."
>
> Response5: We have removed the italicized titles from the contribution list in the revised version.

---

> > ### Author Response · Authors · 2026-01-23
> > **Limitations of Previous vICL Methods in Medical Imaging**
> >
> > Comment1: "“Furthermore, as illustrated in Fig.1.(a) and Fig.1.(c), the masking strategies shown are those adopted during the training phase of MIM-based visual ICL models.” – I read this sentence multiple times to make sense of it. It is a bit broken, I guess. Maybe, rewrite it as “The masking strategies shown in Fig.1.(a) and Fig.1.(c) are those adopted during the training phase of MIM-based visual ICL models.”"
> >
> > Response1: We have adopted your suggestion and revised the sentence in the revised version.
> >
> > Comment2: "“In contrast, the inference stage requires the use of a different masking strategy, as depicted in Fig.1.(d).” – If Fig.1.d is about inference, why does the caption of Fig.1 say it is about training? Please, adjust the figure caption and text accordingly."
> >
> > Response2: Since the masking strategy used in our training is consistent with that used in vICL inference, we have added additional explanations to Figure 1(d).
> >
> > Comment3: "This inconsistency results in a discrepancy between the training and inference procedures, which introduces a gap between training and inference.” – This sentence is also repeating the same thing twice. Yes, there is a discrepancy between training and inference and the paragraph could be refined into a shorter and more effective one."
> >
> > Response3: We have adopted your suggestion and revised the paragraph in the revised version.
> >
> > Comment4: "Standard inference, training-like inference and modified inference: If modified inference is the same as training-like inference but with query label patches replaced with black patches, does it in fact mean “no visible information”? Or if it is as described later (Standard inference vs Modified Inference), please move that description closer to definition. The reader should not search for parts across the manuscript. Also, be consistent with the ordering of items (1,2,3) and their discussion/explanation."
> >
> > Response4: In modified inference, the visible information of the prompt image and query image is fully retained, but the visible information of the query label is completely erased to prevent the model from performing an inpainting task on the query label. We have adopted your suggestion and revised the order of definitions for standard inference, training-like inference, and modified inference in the revised version.
> >
> > Comment5: "“ii) Unnecessarily High Computational Overhead. Although models like Painter and MVG attempt to mitigate this issue by adding image and label patches at shallow layers, they still devote nearly half of their computational budget to prompt processing. This allocation limits the model’s capacity to focus on the query task itself, reducing overall efficiency.” – Who established this? Is there a reference to use here?"
> >
> > Response5: The first sentence is a factual statement, and the second sentence was indeed not rigorous enough. We have removed the second sentence in the revised version.

---

> > ### Author Response · Authors · 2026-01-23
> > **Method**
> >
> > Comment1: "Retrieval-and-Propagate Module for Prompt-Query Fusion: “We introduce a Retrieval-and-Propagate token fusion strategy, which is particularly inspired by the characteristics of medical images from different patients “that” tend to exhibit strong visual similarities within corresponding anatomical regions” – I believe we need “that” or something to bridge two ends there."
> >
> > Response1: We have revised the sentence to make it more fluent.
> >
> > Comment2: "Retrieve and Propagating: do we really need capital letters for these? A bold face should be enough to highlight these keywords used for the proposed method. Same goes for “Copy” baseline."
> >
> > Response2: We have adopted your suggestion and removed the capital letters, using boldface only to emphasize these keywords.
> >
> > Comment3: "What is the Hungarian matching? It needs a little explanation (basic definition and implications). Also, what is the complexity of the Hungarian matching algorithm used here? How does it compare to attention? Is it worth it to abolish one or the other?”
> >
> > Response3: We have included an introduction to Hungarian matching in the appendix, along with a discussion and analysis of its computational complexity, specific runtime, and scalability. Attention is used to measure the relevance between queries. Previously, before employing Hungarian matching, we utilized cosine similarity to measure the relevance between query tokens and prompt tokens. Hungarian matching, on the other hand, aims to complete the pairing between query tokens and prompt tokens based on a similarity matrix. Admittedly, using the vector dot product in attention could also be an alternative metric to the cosine similarity-based approach.
> >
> > Comment4: "“For each query image in the training set, we randomly select another image with the same task and its corresponding label as the prompt image and prompt label.” Random selection of images is an easy but usually a good starting point. As per Appendix, learned prompts were also tested but did not lead to significant improvements. While learned prompts might require further investigation and hyperparameter tuning, could you try a simple kNN-based selection? In the works of Ferber et. al. (2024) in pathology as well as others in ophthalmology, kNN-based selection worked quite well. It would be nice to see its impact here."
> >
> > Response4: We have adopted the prompt selection method from Ferber et al. (2024). The additional results are shown in the following table and have been added to the revised version. Selecting prompts based on an extra visual encoder indeed leads to clear improvements. Other works in natural image vICL have also proposed more complex methods for prompt selection, and we believe this is a worthwhile direction to explore for medical images as well.
> >
> > |                        | Segmentation | Denoising       | Super-Resolution | PET synthesis     |
> > |------------------------|--------------|-----------------|------------------|-------------------|
> > |                        | **Dice**     | **PSNR\SSIM** | **PSNR\SSIM** | **PSNR\SSIM** |
> > | Random Prompt          | 84.95        | 33.01\92.14    | 30.14\91.33    | 31.46\90.17    |
> > | Learned Prompt         | 85.11        | 33.04\92.16    | 30.12\91.32    | 31.39\90.16    |
> > | Selected Prompt[1]        | **85.56**    | **33.29\92.50** | **30.21\91.49** | **32.31\91.32** |
> >
> > [1]Ferber, Dyke, et al. "In-context learning enables multimodal large language models to classify cancer pathology images." Nature Communications 15.1 (2024): 10104.

---

> > ### Author Response · Authors · 2026-01-23
> > **Experiments**
> >
> > Comment1: "By the definition of ICL, we should not change the backbone model’s weights too much. It will be pretty much untouched while we learn from the given context set forth through examples and prompts. So, I assume only the shallow ViT and linear parts of the RandP module were trained for 100 epochs and the rest (large ViT, pixel decoder, etc..) was frozen. If this is the case, it should be made more explicit. Otherwise, would we still call it ICL? If we finetune the whole stack, it would be the old school supervised learning, no?"
> >
> > Response1:We would like to clarify that in-context learning (ICL) does not inherently require the backbone to be frozen during training. The defining property of ICL is that task specification and adaptation are achieved through the provided context (i.e., prompts and examples) at inference time, without parameter updates, making ICL fundamentally an inference-time paradigm rather than a training constraint.
> >
> > While many existing vICL methods adopt frozen or training-free settings in practice, this choice is primarily driven by the availability of large-scale pretrained models and the goal of evaluating zero-shot or prompt-based generalization. For example, backbones pre-trained on large vICL datasets (e.g., LVM [1]) may already exhibit strong zero-shot performance, and recent works such as SD-vICL [2] explore training-free vICL frameworks. These settings are adopted for practical and evaluative reasons, rather than being a defining requirement of ICL.
> >
> > In contrast, our work focuses on the general vICL framework design, following MAE-VQGAN, Painter, PromptGIP, and MVG. We adopt an end-to-end learnable formulation to train the prompt–query fusion module, backbone, and pixel decoder, enabling the model to learn how to exploit contextual information. At inference time, task behavior is determined solely by the provided context without parameter updates, distinguishing our approach from conventional supervised learning where task behavior is fixed in the model weights.
> >
> > In the revised version, we have incorporated the following statement: "Following previous vICL frameworks (Wang et al., 2023; Ren et al., 2024; Liu et al., 2024), we adopt the identical experimental protocol, wherein the prompt–query fusion module, the backbone network, and the pixel decoder are all trained in an end-to-end manner."
> >
> > Comment2: "Table 5 shows First-stage, Multi-stage and Bottleneck; however, in the text, they are given as First, Bottleneck and Multi. Please, be consistent."
> >
> > Response2: We have adopted your suggestion and made the necessary revisions in the revised version.
> >
> > Comment3: "“Experiments show that bottleneck fusion achieves the best performance.” – Yes, it is what we see from the table. Could you elaborate on why this is the case? Maybe, provide a little discussion or speculation to somehow make the reader contemplate or gain insights."
> >
> > Response3: We hypothesize that this is because early stages of U-Net primarily focus on low-level features such as textures and edges[3], where introducing prompt-related information provides limited benefit. In contrast, the role of prompts is more aligned with guiding high-level semantic understanding, for example by indicating the task being performed[4] and providing task-specific insights[5]. Injecting prompt-related information at fine spatial resolutions may therefore be suboptimal, as the prompt inevitably differs from the query image in low-level details. Since the model output is solely optimized for understanding the query image, introducing fine-grained prompt features at early stages may even be detrimental to query interpretation.
> >
> > In the revised version, we have also supplemented a discussion regarding why bottleneck fusion achieves the best performance.
> >
> > [1]Yutong Bai, Xinyang Geng, Karttikeya Mangalam, Amir Bar, Alan L Yuille, Trevor Darrell,
> > Jitendra Malik, and Alexei A Efros. Sequential modeling enables scalable learning for
> > large vision models. In Proceedings of the IEEE/CVF Conference on Computer Vision
> > and Pattern Recognition, pages 22861–22872, 2024.
> >
> > [2]Oorloff, Trevine, et al. "Stable diffusion models are secretly good at visual in-context learning." Proceedings of the IEEE/CVF International Conference on Computer Vision. 2025.
> >
> > [3]Horwath, James P., et al. "Understanding important features of deep learning models for segmentation of high-resolution transmission electron microscopy images." npj Computational Materials 6.1 (2020): 108.
> >
> > [4]Alberto Hojel, Yutong Bai, Trevor Darrell, Amir Globerson, and Amir Bar. Finding visual
> > task vectors, 2024. URL https://arxiv.org/abs/2404.05729.
> >
> > [5]Seungwook Han, Jinyeop Song, Jeff Gore, and Pulkit Agrawal. Emergence and effectiveness
> > of task vectors in in-context learning: An encoder decoder perspective, 2025. URL https://arxiv.org/abs/2412.12276.

---

> ### Comment · Reviewer_NsvT · 2026-01-25
> **Thoughts on the revised manuscript**
>
> I would like to thank the authors for addressing all of my comments so diligently. The manuscript is much more informative and inspiring now. I need to also express my gratitude to the authors for kindly updating my beliefs of the definition of ICL.
>
> I will be happy to change my preliminary rating from "borderline" to "strong accept" in the light of substantial improvements and potential of RandP for many extensions and downstream applications.
>
> On that note, here are few things to quickly fix before the final version.
>
> - The newly added Fig.1 is never referred to in the text, I believe. If it is not relevant, please, remove the figure. If relevant, properly use it to describe ICL by referring to it.
> - On page 5, just below Table 1, there is a new paragraph with one line: "This inconsistency results in ...". I assume the addition of a paragraph was a mistake. The sentence belongs to the previous paragraph.
> - On page 7, "We introduce a Retrieval-and-Propagate token fusion strategy, which is particularly inspired by the characteristics of medical images from different patients tend to exhibit strong visual similarities within corresponding anatomical region". It seems like the changes regarding this sentence did not make it to the new draft. Please, update.
>
> Thanks for your hard work in short time. I would love to see this work at MIDL 2026.

---

> > ### Author Response · Authors · 2026-01-25
> >
> > Thank you sincerely for your thoughtful and encouraging feedback throughout the review process. We are truly grateful for your recognition of our revisions and for updating your assessment to "strong accept." Your constructive comments have significantly strengthened our manuscript.
> >
> > We have addressed all the quick fixes you mentioned in the final version:
> >
> > Figure 1 reference: We have added proper references to Figure 1 in the text where we describe visual ICL.
> >
> > Paragraph formatting on page 5: We have merged the one-line paragraph ("This inconsistency results in...") with the previous paragraph as intended.
> >
> > Sentence correction on page 7: We have revised the sentence to: "We introduce a Retrieval-and-Propagate token fusion strategy, which is particularly inspired by the characteristics of medical images from different patients that tend to exhibit strong visual similarities within corresponding anatomical regions."
> >
> > We deeply appreciate your time and effort in reviewing our work, your insights have been invaluable in improving the clarity and quality of this manuscript.

---

### Official Review · Reviewer_bDvf · 2026-01-09

**Confidence:** 3
**Preliminary Rating:** 4
**Final Rating:** 5

**Summary:**

The paper presents RandP, a visual in-context learning framework for medical imaging that addresses limitations of prior MIM-based approaches, such as training–inference mismatch and high computational cost. It uses a retrieve-and-propagate mechanism with Hungarian matching to directly associate query regions with relevant prompt information. Importantly, the approach works with standard medical backbones like U-Net and Restormer and achieves significantly faster inference than prior vICL methods.

**Strengths:**

1) Clear identification of limitations in existing vICL methods for medical imaging and proposed Novel Retrieve-and-Propagate module, using Hungarian matching for prompt–query token fusion.
2) The massive jump in efficiency and inference speed is a standout feature, as the authors doubled the speed over the standard MIM approach.
3) Strong empirical evaluation across multiple datasets, modalities, and tasks, including both qualitative and quantitative results.
4) By decoupling the prompt-query fusion from the reconstruction loss of the prompt label, the authors successfully extend vICL to non-transformer architectures (UX-Net, Restormer, SwinUNETR).

**Weaknesses:**

1) The Hungarian matching step introduces non-trivial computational overhead and algorithmic complexity - while overall FLOPs are reduced, the scalability of this matching step for higher-resolution inputs or larger prompt sets is not sufficiently discussed.
2) The evaluation primarily compares against other generalist frameworks. Including comparisons with strong single-task specialist models (e.g., task-specific U-Nets) would help better assess the practical performance gap relative to clinically deployed solutions.

**Detailed Comments:**

Please refer to the Weaknesses section.

**Justification Of Final Rating:**

The authors have provided a comprehensive and detailed response that addresses the majority of my technical concerns. The inclusion of GPU-accelerated benchmarks and a broader empirical evaluation across diverse modalities and architectures effectively demonstrates the framework's efficiency.

**Justification Of The Preliminary Rating:**

The paper proposes a well-motivated and technically sound approach to visual in-context learning for medical imaging, showing clear efficiency and performance gains over prior methods. The RandP design effectively addresses key limitations of existing vICL frameworks and is supported by solid empirical results. Some aspects, such as scalability and broader comparisons, would benefit from further analysis to strengthen the work.

**Questions To Address In The Rebuttal:**

Please refer to the Weaknesses section.

---

> ### Author Response · Authors · 2026-01-23
>
> We are truly grateful for the insightful review and the constructive comments provided. In the following, we provide a detailed response to each comment.
>
> Comment1: "The Hungarian matching step introduces non-trivial computational overhead and algorithmic complexity - while overall FLOPs are reduced, the scalability of this matching step for higher-resolution inputs or larger prompt sets is not sufficiently discussed."
>
> Response1: We thank the reviewer for raising the concern regarding the computational overhead of the Hungarian matching step. We indeed acknowledge that classical Hungarian matching has a cubic time complexity with respect to the sequence length. To make this aspect transparent, we report the execution time of the Hungarian matching step using both the SciPy implementation and the GPU-accelerated version from HA4DeTR [1] under different token sequence lengths. In our current experimental setting, the sequence length is fixed to 256, where the matching cost remains negligible compared to the overall network inference time.
>
> However, the scalability issue highlighted by the reviewer can be effectively mitigated in practice. Specifically, our RandP module performs matching between the query image and each prompt image independently, leading to linear scaling with respect to the number of prompts. More importantly, when moving to higher-resolution inputs, we do not require global Hungarian matching over all tokens. By leveraging anatomical priors, namely, the spatial consistency of corresponding anatomical regions across different patients, Hungarian matching can be restricted to localized spatial windows at high resolution.
>
> | Seq_length | GPU (ms) | SciPy (ms) | ViT (ms) |
> |------------|----------|------------|----------|
> | 256        | 0.125    | 0.57       | 13.8     |
> | 512        | 0.427    | 3.05       | —        |
> | 1024       | 1.607    | 12.93      | —        |
> | 2048       | 8.082    | 54.38      | —        |
>
> This localized matching strategy preserves the benefits of optimal bipartite assignment while avoiding the inefficiencies of greedy matching, and significantly reduces the practical computational burden. Under this design, the matching cost scales linearly with image resolution rather than cubically with the global token count. We have added a concise introduction to Hungarian matching in the appendix, together with a discussion of its computational complexity, runtime analysis, and scalability considerations.
>
> Comment2: "The evaluation primarily compares against other generalist frameworks. Including comparisons with strong single-task specialist models (e.g., task-specific U-Nets) would help better assess the practical performance gap relative to clinically deployed solutions."
>
> Response2: We appreciate the reviewer’s suggestion, and we agree that such comparisons are important. In the table below, we provide a comparison of the performance gap between our RandP model and recent state-of-the-art models specifically designed for prostate segmentation on the PROMISE12 dataset. Additionally, we compare our model’s performance on low-dose CT denoising against recent SOTA models tailored for this task.
> | Dataset   | Model           | Dice  |
> |-----------|-----------------|-------|
> | PROMISE12 | RandP           | 89.35 |
> | PROMISE12 | KanGMSA-Net [2] | 92.49 |
>
> | Dataset     | Model           | PSNR/SSIM    |
> |-------------|-----------------|----------------|
> | AAPM-Mayo   | RandP           | 33.0/92.1      |
> | AAPM-Mayo   | MRED-Net [3]    | 33.5/94.6      |
>
> While we acknowledge that specialized models for specific tasks still outperform our approach, we believe that vICL, through large-scale pre-training and careful prompt selection for downstream tasks, can narrow this performance gap and offer significant value in practical clinical applications.
>
> [1]Feng Lin, Xiaotian Yu, and Rong Xiao. GPU-Accelerated Batched Hungarian Algorithm for DETR. GitHub repository, 2025. https://github.com/linfeng93/HA4DETR
>
> [2]Li, Chunyu, et al. "KanGMSA-Net: A novel prostate segmentation framework integrating Kolmogorov–Arnold networks and grouped multi-scale attention." Biomedical Signal Processing and Control 113 (2026): 109081.
>
> [3]Liu, Yuhang, et al. "MRED-Net: A pure tokens-to-token visual mamba-based residual encoder-decoder network for low-dose CT denoising." Expert Systems with Applications 301 (2026): 130305.

---

### Author Rebuttal · Authors · 2026-01-23

**Rebuttal:**

Here is our revised version.

**Supporting Material:**

/attachment/e6cf20b64a2e18406ebd97cfc04790da4f6a257e.pdf

---

### Meta-Review · Area_Chair_DzCA · 2026-02-03

**Recommendation:** Accept (Oral)
**Confidence:** 4

**Metareview:**

The rebuttal well clarified the concerns; all reviewers agreed that the contribution of this work is solid and should be accepted. The AC agreed with that and suggested acceptance.

---

### Decision · Program_Chairs · 2026-02-13

Accept (Poster)